# Regulation of Cell Plasticity by Bromodomain and Extraterminal Domain (BET) Proteins: A New Perspective in Glioblastoma Therapy

**DOI:** 10.3390/ijms24065665

**Published:** 2023-03-16

**Authors:** Deborah Gargano, Marco Segatto, Sabrina Di Bartolomeo

**Affiliations:** Department of Biosciences and Territory, University of Molise, I-86090 Pesche, Italy

**Keywords:** BET proteins, cell plasticity, glioblastoma, differentiation, epigenome, reprogramming therapy

## Abstract

BET proteins are a family of multifunctional epigenetic readers, mainly involved in transcriptional regulation through chromatin modelling. Transcriptome handling ability of BET proteins suggests a key role in the modulation of cell plasticity, both in fate decision and in lineage commitment during embryonic development and in pathogenic conditions, including cancerogenesis. Glioblastoma is the most aggressive form of glioma, characterized by a very poor prognosis despite the application of a multimodal therapy. Recently, new insights are emerging about the glioblastoma cellular origin, leading to the hypothesis that several putative mechanisms occur during gliomagenesis. Interestingly, epigenome dysregulation associated with loss of cellular identity and functions are emerging as crucial features of glioblastoma pathogenesis. Therefore, the emerging roles of BET protein in glioblastoma onco-biology and the compelling demand for more effective therapeutic strategies suggest that BET family members could be promising targets for translational breakthroughs in glioblastoma treatment. Primarily, “Reprogramming Therapy”, which is aimed at reverting the malignant phenotype, is now considered a promising strategy for GBM therapy.

## 1. Introduction

Glioblastoma multiforme (GBM) is the most common and aggressive malignant primary brain tumor in adult humans, characterized by a poor prognosis despite the existence of multimodal therapy [1]. Indeed, in the last 30 years a limited progress has been made in GBM treatment with current first-line standards-of-care involving maximal safe tumor resection with adjuvant temozolomide (TMZ)-based chemotherapy and radiotherapy [2]. GBM response to these therapeutic approaches is limited due to (1) the broad tumor capability to invade the surrounding brain tissue, that makes it unrealistic for surgery to remove all tumor cells; (2) the presence of a sub-population of drug-resistant cells. Altogether, these factors make inevitable relapses occurrence [3,4]. In detail, GBM recurrence arises from residual cells at the margin of the resection that rapidly acquire radio- and chemo-resistance during treatment and cannot be efficiently counteracted by any treatments [5]. Moreover, resistant GBM cells are not easily accessible to chemo-therapeutic drugs due to the presence of the blood-brain-barrier (BBB) that prevents drugs from reaching the tumor [6]. Therefore, it appears evident that there is a compelling need for novel and more effective treatment strategies [7]. Recently, epigenetic therapy has gained increasing interest, consisting of the manipulation of the cancer epigenome through the targeting of epigenetic factors frequently deregulated in several type of malignancies. Genes and proteins that control epigenetic alterations have become potential novel therapeutic targets for tumors, particularly attractive due to the reversibility of epigenetic modifications. 

The epigenetic landscape of GBM has been thoroughly explored and many epigenetic alterations, such as histone modification, DNA methylation, and chromatin remodeling, have been mechanistically linked to the biological features of the tumor. Among these, the chromatin readers bromodomain and extra-terminal domain (BET) proteins were found to be significantly overexpressed in GBM tissue rather than in the normal brain [8]. BETs belong to an evolutionarily maintained protein family [9,10], that regulates gene expression via recognizing acetylated lysine residues on histone and nonhistone chromatin factors [11,12]. Deregulation of BET expression leads to changes in transcriptome with consequent loss of cellular identity and proper functioning, the main driving-force of cancerogenesis. Therefore, their altered expression in GBM and the therapeutic effectiveness of pharmacological inhibitors, observed in a wide range of cancers, including hematopoietic malignancies [13], neuroblastomas [14], breast cancer [15], and prostate cancer [16], have suggested BET proteins as promising targets for the treatment of GBM. In this review, we discuss the role of the epigenetic readers in the regulation of cellular identity in physiological and tumorigenic contexts, and the putative translational breakthroughs in glioma treatment. 

## 2. Cell Plasticity in Cancer Onset

Tumorigenesis is a dynamic process which takes place in several steps that drive the progressive transformation of normal human cells into their highly malignant counterparts [17,18,19]. In this regard, tumor development is formally compared to Darwinian evolution in which a succession of genetic changes leads to the onset of hallmarks of cancer cells [20]. During the earliest stages of neoplastic transformation, the occurrence of an epigenetic reprogramming causes genomic instability and subsequent transcriptome alteration, leading differentiated cells to re-acquire cell plasticity. Indeed, it is well established that a critical component of cancer pathogenesis is the progressive loss of cellular identity and the evasion from a differentiated state [21]. Recently, Hanahan et al. have summarized the three possible mechanisms through which cancer cells could reverse their fate and re-acquire plasticity [22]. According to this model, cancer cells can originate from completely differentiated cells which undergo a de-differentiation step to progenitor-like cell state (de-differentiation hypothesis). Conversely, based on the “arrested differentiation hypothesis”, a neoplastic cell could arise from a progenitor cell that is destined for end-stage differentiation but short-circuits the process, maintaining the cell in a partially differentiated, progenitor-like state. Alternatively, a trans-differentiation mechanism may operate, in which cells that were initially committed to a specific differentiation pathway switch to a completely different developmental program, thereby acquiring tissue-specific traits that were not preordained by their normal cells-of-origin. The identification of the cellular origin of a malignancy would help to understand the mechanisms of tumor initiation/propagation and to identify specific molecular targets eligible to counteract cancer cells. Noteworthy, a distinction should be made between cell-of-origin, also referred to as tumor-initiating-cell, and cancer stem cell (CSC). “Cell-of-origin” refers to normal cells in which oncogenic mutations occur and accumulate, thus starting the tumor formation process, while CSC refers to a subset of proliferating cancer cells that sustain tumor growth [23]. 

### 2.1. Glioma Stem Cells (GSCs)

The GSC hypothesis arises from the evidence that a hallmark of glioblastoma is the high degree of morphological, molecular, and cellular heterogeneity, due to the presence of distinct cellular subpopulations harboring potent tumor-initiating capability. Almost two decades ago, Galli and colleagues characterized Glioma Stem Cells (GSCs) discovering that, unlike other brain cancers, GBM consists of transformed precursors sharing a full complement of functional features with their non-tumoral neural stem cells counterpart [24]. They found that the GBM specimens-derived cells were able to form tumors reproducing the main histologic, cytologic, and architectural features of the original mass. Moreover, GSCs also showed unipotency (toward astroglial fate) in vivo and multipotency in vitro (toward neuronal-astroglial-oligodendroglial fate) [24]. Further studies clarified common and uncommon key functional features and regulatory cues of GSCs and normal Neural Stem cells (NSCs) [25,26]. Both GSCs and NSCs often express stem cell molecular biomarkers, such as CD133, SOX2, and NOTCH1; furthermore, both kinds of cells share the ability to form neurospheres in serum-free conditions and to form tumors in orthotopic xenograft animal models [27,28,29]. Moreover, proliferation and fate of both NCSs and GSCs are influenced by the same growth factors, cytokines and chemokines [30,31]. However, dysregulation of NCSs-relevant signaling pathways, such as Sonic Hedgehog, Notch, Wnt, Bmi-1, is believed to drive GCSs development [32,33,34,35,36,37,38]. This evidence was consistently correlated to the phenotypic and genomic alterations found in patient tumors [39,40]. Therefore, a GSCs hypothesis has been proposed, that incorporates a model in which these stem-like cells are the apex of a dynamic cellular hierarchy, where, on one hand, they self-renew to replenish themselves and, on the other hand, they give rise to a tumor cell progeny with a more restricted cell plasticity, thus generating tumor heterogeneity [41]. Due to the upregulated multi-drug transporters, an altered anti-apoptotic machinery, and an enhanced DNA damage response, GSCs are relatively resistant to both chemotherapy and radiotherapy [42], substantially contributing to tumor spreading and recurrence. Therefore, GSCs represent a golden cellular target in glioblastoma treatment [43,44,45], aimed at rescuing the signaling pathways that underpin their striking tumorigenic and invasive capability [46,47]. Very recently, GSCs hypothesis has been revised. Trivieri and collaborators discovered a subset of GSCs, derived from high-grade gliomas, characterized by mitogen-independent growth capability, coexisting with the classical GSCs [48]. Overexpression of Wnt5a and a downregulation of EphA2 were related to the enhanced aggressiveness of growth factor-independent GSCs compared to the classical growth factor-dependent GSCs [48]. The coexistence of both GSCs types inside the same tumoral mass and the possibility to convert one GSCs type into another, by manipulation of exogenous mitogens exposure, suggested a lineage relationship between the two distinct GSCs populations. According to Trivieri’s hypothesis, a specific environmental cue is a determinant of GSCs “oscillatory” highly invasive/less proliferative behavior and a driver of GBM heterogeneity, that make GSCs so difficult to eradicate [48]. Nevertheless, the cell type determining GBM onset remain uncertain. 

### 2.2. The “Cell-of-Origin” of GBM

The comprehension of the origin of GBM cells is crucial to the understanding of tumor biology, and also to developing novel therapeutic strategies [49]. Indeed, a major challenge in drug discovery is to eliminate the cells responsible for tumor onset. However, GBM initiating cells continue to be a source of debate. According to the three possible mechanisms described above, GBM initiating cells may originate from neural stem cells (NSC), from pluripotence-restricted astrocyte-like stem cells, or from terminally differentiated glial cells (Figure 1).

#### 2.2.1. De-Differentiation Hypothesis

According to this model, putative cells-of-origin of GBM are mature astrocytes, which undergo a de-differentiation process into a neural-stem-like state (Figure 1) [50]. Indeed, animal models with loss-of-function mutation in TP53, PTEN, and/or RB1, as well as cortical astrocytes silenced for NF1 and TP53 or Ras and TP53, result in glioma formation [51,52]. These genomic alterations induce astrocytes to de-differentiate into neural progenitor-like state, in which the transcriptional factors Sox2, c-myc and Nanog are re-expressed. It has been shown that loss of INK4A/Arf also induces astrocytes to de-differentiate and to undertake the malignant transformation via activation of K-Ras and Akt [53]. Otherwise, the combined loss of tumor suppressors p16(INK4a) and p19(ARF) led to astrocytes de-differentiation in response to a constitutive EGFR activation [54]. Lastly, it has been demonstrated that loss of p53 combined with overexpression of oncogenes such as Akt and c-Myc not only induces gliomagenesis, but also induce the expression of stemness markers in mature astrocytes [55,56].

#### 2.2.2. “Arrested Differentiation” Hypothesis

In a developing Central Nervous System (CNS), NSCs are ubiquitous and responsible for the formation of the three main nerve tissue cell types [57]. NSCs progressively restrict their cell plasticity, giving rise to intermediate progenitor cells that can ultimately differentiate into mature neurons or glial cells (Figure 1). A subset of NSCs and lineage-restricted progenitor cells continue to reside in restricted regions of the postnatal and adult brain: the subventricular zone (SVZ) of the lateral ventricle and the subgranular zone (SGZ) of the dentate gyrus (DG) in the hippocampus [58,59]. The human SVZ consists of three anatomically distinct layers: the ependymal layer, hypocellular gap, and astrocytic ribbon [60]. The latter one contains glial fibrillary acid protein (GFAP)-positive cells and astrocyte-like stem cells with a restricted self-renewal potential, compared to NSCs, whose fate and terminal differentiation are rapidly achieved [59,61]. It has been hypothesized that during the earliest stages of gliomagenesis, the occurrence of a mutation on TERT (Telomerase Reverse Transcriptase) promoter triggers high grade malignant transformation in lineage-restricted astrocyte-like stem cells from the astrocytic ribbon layer [61]. Subsequently, several other somatic mutations occur and accumulate [62]. It has been shown that the occurrence of the most common alterations in deregulated oncogenes in GBM (EGFR, PTEN, TP53, NF1) allows mutated progenitor cells to migrate from SVZ to the cortex, leading to the growth of a tumor [61,63]. 

#### 2.2.3. “Circumvented Differentiation” Hypothesis

It has been also suggested that GBM originates from a population of progenitor cells, localized in the outer part of the primate SVZ referred to as radial glial cells (Figure 1) and more specifically, from the outer radial glial cells (oRG) [64,65]. Indeed, during neurogenesis, neuroepithelial cells transform into a different population of progenitor cells, called radial glial cells, which are characterized by the expression of astroglial markers, such as the glutamate transporter (GLAST) or the glial fibrillary acidic protein (GFAP) and the lipid binding protein (BLBP) [66]. Radial glial cells divide asymmetrically to maintain the progenitor pool and to produce a neuronal-committed daughter cell [67,68,69,70]. The cell committed to the neuronal lineage becomes either a neuron (direct neurogenesis) or an intermediate progenitor that undergoes another division before leaving the proliferative area (indirect neurogenesis). As the residual radial glial cells become astrocytes during development, radial glial cells are not expected to be present in the normal adult CNS [71,72,73,74]. Therefore, it has been proposed that a pool of latent or quiescent radial glial cells exits the cell cycle and becomes dormant, thus maintaining proliferative potential and giving rise to glioma initiating cells and glioma [64,75]. The phenotypic similarity between any GBM cells and radial glial cells can be considered as an argument for the origin of GBM [76]. Indeed, GBM cells co-express SOX2 and Nestin, but also GFAP [77]. This is a typical phenotype for GFAP-positive neural progenitors or radial glia but not for classical NSCs or glial precursors [64,78]. However, GFAP expression is a hallmark of the astrocyte lineage commitment and the loss of GFAP is essential for cells such as NPs or radial glial cells to differentiate into neurons or into oligodendrocyte [79,80]. GBM cells show a differentiation pathway similar to that observed in radial glial or in any GFAP-positive progenitors and different to the classical NSC one. The differentiation process appears to be blocked at early stages in GBM cells [77,81,82]. Moreover, glioblastoma-derived primary cells undergo mitotic somal translocation, a characteristic mode of division of radial glial cells occurring during human development, in which the soma translocates toward the cortical plate prior to cytokinesis [64]. Overall, these findings suggest that reactivation of developmental programs typical of radial glial cells occur in GBM cells. The above described phenotypical similarities between GBM cells and GFAP-positive cells can be essential, when designing therapies for GBM patients. 

## 3. Bromodomain and Extra-Terminal Domain (BET) Proteins

Bromodomain and extra-terminal domain (BET) proteins are a class of epigenetic protein called “chromatin readers”, whose function is to recognize acetylated lysine on histone tails and promote downstream signals [83]. In the human genome 61 types of bromodomains are known, phylogenetically grouped in eight major protein families, and BET are comprised into the second one [84]. More recently, a function-based classification has been proposed, providing a more comprehensive approach than the previous one based on protein structures. According to this classification, BET proteins have been placed in the subgroup V of transcriptional co-activators [85]. The BET family consists of four evolutionarily well conserved members, including BRD2, BRD3, and BRD4, which are ubiquitously expressed in all tissues, and the testis-specific BRDT. The last-mentioned one is expressed only in male germline cells where it plays a key role in spermatogenesis [10,11]. All members share the same three-dimensional structure characterized by two tandem amino-terminal bromodomains, BD1 and BD2 and an extra-terminal domain (ET) at carboxy-terminus [10,11,12]. The two bromodomains BD1 and BD2 consist of a sequence of 110 amino acids structured in left-handed four-helix bundle (αZ, αA, αB, αC), interposed by two inter-helical loops (ZA and BC). These helical modules create the hydrophobic binding pocket for the selective recognition of acetylated lysine residues on N-terminal histone tails and other proteins, which is essential for chromatin interaction [86,87]. A comparative analysis revealed that overall sequence identity between BD1 and BD2 is only 35–45%, while the substrate binding pocket is perfectly conserved, with a 100% sequence overlap, in all eight BET BD domains [88]. Nevertheless, sequence and structural alignments of the BET bromodomains have highlighted five highly variable positions close to the acetylated lysine binding pocket. Therefore, these variable positions have been suggested to be critical for individual BET member specific interactions [88]. Due to the high degree of structural conservation, potent and selective BET inhibitors for the individual bromodomains are mostly lacking, hindering the understanding of the specific biological role of each bromodomain. Moreover, the lack of selective inhibition also explains some of the off-target effects and/or safety concerns observed with this class of molecules. However, more recently, the distinct roles of BET-BD1 and BET-BD2 in chromatin interaction have been clarified. BD1-chromatin binding is required to maintain the established gene expression program. Instead, after a given stimulus, BD2 facilitates the stable recruitment of BET proteins on the chromatin, thus eliciting a rapid change in gene expression [89]. Similarly to the BET bromodomains, the ET region is a well conserved sequence of almost 80 amino acids structured in a protein-protein interaction domain responsible for chromatinic recruitment of transcriptional co-activators [90,91,92,93,94,95]. It is known that the distinct function of BET proteins is critically determined by differences in the C-terminal domain, whereas the N-terminal domain is mainly responsible for protein stability. Specifically, the four BET members differ in an extended ET region containing a coiled-coils (CC) structural motif [96]. Indeed, the adjacent CC region is found only in BRD2 and BRD3, but not in BRD4. Interestingly, the CC motif does not act as an independent protein-docking site, but works in close association with the neighbouring ET domain. Indeed, the CC region functions as a molecular scaffold for PAF (RNA polymerase II-associated factor) and CK2 (casein kinase II) complexes [96], which are known to occupy active chromatin [97] and associate with elongation factors [98]. Instead, BRD4 can exist as a long isoform containing an extended C-terminal tail consisting of a conserved region of almost 40 amino acids needed for the recruitment of the positive transcription elongation factor (P-TEFb), required for the RNA polymerase II activity [99,100,101]. The short isoform of BRD4 lacks this C-terminal domain and is therefore structurally similar to BRD2 and BRD3. However its expression and function in most cell types remains poorly characterized [102,103]. The evolutionary preserved structural features make it difficult to understand whether different BET bromodomains contribute to cell- or gene-specific functions, leading to an incomplete comprehension of the high degree of functional redundance among BET members and their isoforms. Although the interactions with the other proteins appear to differ among BET members, it is well established that BRD2, BRD3, and BRD4 co-localize for approximately two thirds of their chromatin binding sites, suggesting their broadly shared regulatory function [104]. 

BET proteins are involved in a variety of biological processes, such as chromosomal architecture [105], DNA replication [106], and DNA damage repair, but are best known for their role in transcriptional regulation [107,108]. BETs mechanistically link chromatin status to gene transcription “reading or “interpreting” acetylated chromatin and recruiting transcriptional regulatory complexes to their binding sites [12,109]. Thus, BET proteins mainly function as molecular scaffolds that guide the assembly of nuclear macromolecular complexes to specific sites on chromatin in order to provide the dynamics of gene expression [110,111].

### BET Protein Inhibitors

BET bromodomain inhibitors (iBETs) are successfully employed in the treatment of inflammatory, metabolic, and cardiac disorders, as well as in cancer therapy [12,112,113,114,115]. Since their first identification almost a decade ago, iBETs have attracted considerable interest as anti-tumoral agents, due to their ability to induce an efficient downregulation of c-Myc expression [114,116,117,118]. c-Myc oncogene is involved in driving several human cancers, particularly haematopoietic malignancies, and has long been considered “orphan” with respect to pharmacological inhibition. The central role of BET proteins in the pathogenesis of a variety of human diseases, including cancers, but also heart failure, viral, autoimmune, and inflammatory diseases, has greatly accelerated the discovery of a number of small molecules, capable of selectively inhibiting the BET bromodomains [12,110,119,120,121,122,123]. iBETs induce a global remodeling of protein interactions around BET proteins and regulate downstream cellular responses through a competitive binding to the substrate binding pocket [124]. These small molecules target bromodomains with higher affinity, displacing the epigenetic enzymes from their association with acetylated lysine residues and co-factors on chromatin [114,116]. Most of the iBETs currently available, referred to as pan-BET inhibitors, indistinctively bind both BDs of all four paralogues and are unable to selectively discriminate among the BET members [114,116,125,126]. Therefore, predicting a gene response to pharmacologic BET inhibition remains a challenge, probably because cells express several BET members with chromatin occupancy overlapping [96]. This lack of selectivity is the main limitation in iBETs clinical administration so far. Moreover, the widespread side effects that have been reported, such as thrombocytopenia, testis toxicity and gastrointestinal toxicity, devalue the full potential of their action [89]. It has been proposed that the adverse events of pan-BET inhibitors are due to their interference with BET functions in tissues other than the target ones [89]. For example, testicular toxicity could be caused by the interference with BRDT function during spermatogenesis [127,128,129]. Similarly, thrombocytopenia may result from the inhibition of BRD3 during erythroid cell maturation, a process in which BRD3 has been shown to be involved [130]. However, several iBETs have entered clinical trial. For example, OTX015 (MK-8628) has shown a robust anti-tumoral activity in in vitro and in vivo pre-clinical models of a wide range of human tumors, including haematopoietic malignancies, neuroblastoma, breast, and prostate cancer [17,131] and then it has progressed to Phase I of clinical studies against GBM and medulloblastoma (https://clinicaltrials.gov (accessed on 26 January 2021), NCT 02296476). In addition to its ability to cross the blood-brain barrier, OTX015 has also demonstrated specific selectivity for tumor/GBM tissue compared to the surrounding brain tissue [132,133]. These data provide a solid pharmacological basis for the potential interest of OTX015 as therapeutical agent for the treatment of brain cancers. 

JQ1 is one of the first iBETs described and so far it is considered a promising candidate in preclinical studies, due to its excellent oral bioavailability, good pharmacokinetics, and ability to cross the blood-brain barrier [10]. However, its short half-life, of almost 1 h in vivo, has strongly limited its application in the clinical practice [116]. 

In recent years, the efforts of drug discovery have focused on identifying selective inhibitors for each member of the BET family. To reduce toxicity and maintain a durable inhibitory effect, the first class of iBETs has been replaced by a more selective second generation of iBETs. In particular, a structure-based design allowed to generate either BD1 or BD2-selective molecules, with little binding capacity to bromodomains not belonging to the BET family; specifically, GSK778 and GSK046 are referred to as iBET-BD1 and iBET-BD2, respectively [89]. In 2016, a bivalent BETi, named AZD5153, was optimized to interact with both the BRD4 bromodomains [134]. In the following years a few BRD4-selective inhibitors have been described. The molecule ZL05880 is a weak BRD4 BD1-biased inhibitor (IC50 = 5 mM) that also shows detectable binding to BD2s [135]. The inhibitor FL-411 preferentially binds the two BDs of BRD4 with no activity against the other BDs, with the exception of BRD2-BD1. Moreover, ZL0420 and ZL045, that have been described as nanomolar BRD4 BDs inhibitors, also show significant efficacy against other BET BDs [136]. Based on the proteolysis targeting chimera (PROTAC) technology, a new generation of iBETs has emerged, which interferes with the function of epigenetic readers by promoting their proteasomal degradation [137]. Indeed, the PROTAC inhibitors act as BET degraders that direct the specific BET member to proteasomal degradation, working as molecular bridges between a specific BET member and E3 ubiquitin ligase [138]. An example of PROTAC iBET are MZ1 and its derivative AT1 which show selective degradation of BRD4 versus BRD3 and BRD2, while ZXH-3-26 demonstrates the same profile using CRBN as E3 ligase [137,139,140]. Surprisingly, with a molecular mechanism completely unknown, an analogue of (+)-JQ1, named GNE0011, has been reported to trigger proteasomal and ubiquitin-dependent selective degradation of BRD4 over BRD2 and BRD3, with very similar efficacy of the PROTAC molecules [141].

In light of this evidence, the possibility of selectively inhibiting the two bromodomains of the BET members would allow us to find answers to some crucial questions about this family of proteins. For example, it would be interesting to understand why most cells express all the three BET proteins, and why the two BET domains in all four paralogues share a high degree of structural conservation, showing a similar preference for the binding to deacetylated tails of histones [89].

## 4. BET Proteins Regulate Progenitor Commitment during Mammalian Embryonic Development

During development, embryonic/pluripotent stem cells (ESCs/PSCs) can differentiate into any cell type within the adult body, including terminally differentiated cells such as pancreatic, liver, neural and cardiac cells [142]. The epigenetic factors involved in lineage commitment represent effective targets for translational breakthroughs in therapeutics. The epigenetic regulation plays a critical role in directing the transcriptome towards stem cell differentiation and lineage commitment. A hallmark of stem cell differentiation is a directional switch from a transcriptionally permissive euchromatin state to a well-established heterochromatin state [143]. Changes in histone marks control chromatin remodeling resulting in a finely regulated transcriptome landscape in which only a specific set of genes remains transcriptionally active [144]. Therefore, cell lineage specification is ensured by a well-coordinated interplay between histone “writers”, such as histone acetyltransferases and histone methyltransferases, and histone “erasers” such as histone deacetylases and demethylases [145,146]. Although developmental genetics and embryology provided much insight into the signaling pathways controlling differentiation, the molecular events that coordinate the exit from pluripotency and differentiation steps are poorly understood [147]. 

BET proteins play several biological functions. Among them, the regulation of gene expression and chromatin organization are the best characterized ones [148,149]. In accordance with the essential role of lysine acetylation in modulating gene expression and cell plasticity, a growing body of evidence demonstrates the pivotal role played by BET proteins in cell fate decision during mammalian development (Table 1). Notably, it has been shown that a fine interplay between BRD4 and BRD2 is crucial for the specification of mouse pluripotent embryonic stem cells (mESCs) [150]. BRD4, indeed, maintains mESCs pluripotency by suppressing autocrine Activine/Nodal production through the binding of multiple Nodal gene regulatory elements (NREs) at the Nodal promoter. NREs contain bromodomain binding motifs on lysine-acetylated histone H4 [116], which are responsible for the direct recruitment of BET family members [125]. BRD switching at Nodal promoter elements occurs with increased BRD2 occupancy at NREs following BRD4 displacement. The result is a Nodal-Smad2- dependent mesendoderm differentiation with coordinated mESCs pluripotency exit [150]. In human ESCs, BRD4 has been also demonstrated to suppress neuroectodermal lineage commitment by indirectly repressing the neuroectodermal differentiation genes [151]. Concurrently, BRD4 upregulates the expression of genes known to directly repress differentiation, resulting in the maintenance of hESCs pluripotency [152,153]. Coherently, BET inhibition leads to an increased expression of the neuroectodermal lineage marker (NeuN) due to a BET-dependent Oct4 downregulation [151]. In the neonatal mouse neonatal brain, BRD2 is the most abundant BET member expressed during oligodendrocyte differentiation. Indeed, it has been found that BET proteins regulate the cell fate of oligodendrocytes progenitors and their expression levels progressively decline in developing white matter tracts [154]. Accordingly, BET-BD1 selective inhibition facilitates the differentiation of primary oligodendrocytes progenitors, whereas BET-BD2 or pan-BET inhibition negatively affects the differentiation process, demonstrating that BET proteins play a central role in axons myelination during the embryonic development [154]. BET proteins are also involved in the specification of neural stem cells (NSCs) cell fate specification. As for primary oligodendrocytes progenitors, the expression of BETs markedly decrease during the differentiation process [155]. In addition, genetic and pharmacological inhibition of BET proteins promote a well-established transcriptional program that induces NSCs toward a neuronal fate rather than glial fate. Taken together these data suggest that BET chromatin readers are pivotal in neural cell fate decision during CNS development [155]. 

Recently, BET proteins have also been involved in the regulation of melanocytes differentiation (Table 1). BRD4 and BRD2, but not BRD3, induce differentiation of unpigmented melanoblasts via a physical interaction with MITF (melanocyte inducing transcription factor) the master regulator of melanocyte development. BRD4 and BRD2 act as molecular scaffolds at promoters surrounding MITF-binding sites of TYR and TYRP1 genes, encoding enzymes required for melanin synthesis [156]. Therefore, BET pharmacological inhibition or BRD4/BRD2 silencing inhibits the expression of the two melanin synthesis enzymes TYR and TYRP1, impairing melanocyte differentiation and melanin synthesis. 

Myogenic differentiation also consists of a sequential genes activation, that is tightly regulated by the interplay between myogenic transcription factors and epigenetic changes [157,158,159]. BET family drives skeletal myogenesis through a dynamic expression pattern during the transition from myoblast to myotubes (Table 1). In the first stage of differentiation process BRD2 is highly expressed and is stabilized at lower levels at later stages. In contrast, BRD3 is expressed at very low levels in myoblasts, but its protein levels are sustained during differentiation to myotubes. On the other hand, BRD4 expression increases to high levels and then gradually declines as differentiation progresses [160]. Nevertheless, the three members exert reciprocal regulatory effects on myogenic differentiation by binding to the H3K27me3 enriched *Myog* promoter, with BRD3 inhibiting and BRD4 inducing skeletal myogenesis [160]. 

BRD4 also transduces the transcription program promoting adipocyte differentiation in mice, driving the expression of Pparg and Cebpa transcription factors, both required for adipocyte differentiation [161]. Moreover, both BRD3 and BRD4 bind to the acetylated GATA1, the erythroid master transcription factor that is pivotal in the activation of all erythroid-specific genes as well as in the downregulation of progenitor genes [130,162,163,164]. Similarly to what has been observed in melanocyte differentiation, BET members bind to the largest number of GATA1-occupied sites, thus promoting the expression of the transcriptional program required for hematopoietic maturation [130].

**Table 1 ijms-24-05665-t001:** BET proteins involvement in Cell Plasticity.

Progenitor Cells	BET Member	Role on Cell Fate	Mechanism	References
mESCs	BRD4	Pluripotency maintenance	Activin/Nodal inhibition	[150]
	BRD2	Mesendoderm specification	Nodal-Smad2 induction	
hESCs	BRD4	Neuroectodermal suppression	Stem Cell gene induction (i.e., Oct4)	[151]
Oligodendrocytes progenitors	BRD2	Stemness maintenance	Unknown	[154]
NSCs	BRD2/BRD3/BRD4	Neuronal fate suppression	Unknown	[155]
Unpigmented Melanoblasts	BRD2/BRD4	Melanocyte differentiation	TYR and TYRP1 induction via MITF interaction	[156]
Myoblasts	BRD3 BRD4	Skeletal myogenesis suppression Skeletal myogenesis induction	*Myog* induction	[160]
Adipoblasts	BRD4	Adipocytes differentiation	Pparg and Cebpa induction	[161]
Erythoblasts	BRD3/BRD4	Erythroid maturation	GATA1 binding	[96,130,162]

## 5. Role of BET Proteins in GBM Biology

### 5.1. BET Proteins in Tumor Biology

A critical component of cancer pathogenesis is epigenome dysregulation. Genome wide sequencing of human cancers revealed that most cancers harbor frequent mutations in genes encoding for components of the epigenetic machinery, and almost half of human cancers harbor mutations in genes encoding for proteins that structure chromatin [165,166,167,168]. The results of these mutations are abnormalities in the epigenome, deregulation of gene expression, and decreased genomic stability [166,168]. A classic example of epigenetic dysregulation is the accumulation of transcriptionally activating lysine acetylation at enhancer regions of oncogenes such as c-Myc [169]. Thus, BET proteins are directly involved in the transcriptional activation of oncogenes through their recruitment to hyper-acetylated regulatory regions [170]. The oncogenic role of BET proteins was first discovered in NUT midline carcinoma (NMC) [171,172,173]. NMC is characterized by the t(15;19) translocation of BRD3 and/or BRD4 genes, leading to a fusion protein with nuclear protein in the testis (NUT). BRD3/4-NUT oncoprotein drives the onset of NMC by maintaining MYC expression and blocking the differentiation of midline body structures [173,174]. In addition to NMC, the contribution of BET proteins to cancer progression has been extensively reported in 20 other types of common cancers [175]. Indeed, dysregulation of BET proteins promotes an aberrant chromatin structuring in several types of cancers such as AML (acute myeloid leukemia), BL (Burkitt lymphoma), MM (multiple myeloma), etc. [117,176,177]. For instance, prostate cancer development, is driven by BETs overexpression, which induces chromatin decondansation, an aberrant androgen receptor activation, and c-myc overactivation [178,179]. In breast cancer, BET members differentially regulate an aggressive phenotype with BRD2 as a positive regulator of epithelial-to-mesenchymal transition (EMT) and BRD3 and BRD4 as repressors of this program [180]. BRD4 has also been shown to regulate invasion and migration in cellular models of triple negative breast cancer via the Jagged1/Notch signaling pathway [181]. 

BETs have also been implicated in neuro-oncology [182,183]. The negative effects achieved by iBETs on the main oncogenic properties of ependymoma, medulloblastoma, neuroblastoma, and GBM allow BET proteins to be referred to as brain cancer drivers. 

### 5.2. BET and Glioma

Epigenetic modifications exert a crucial role in gliomas onset and progression [184,185,186]. To date, many researchers have highlighted a key role of several epigenetic phenomena in GBM tumorigenesis, such as DNA methylation, histone modifications, and chromatin remodeling [187]. Alterations in DNA methylation observed in GBM include genome-wide hypomethylation, gene-specific hypomethylation and promoter-specific hypermethylation [188], all corresponding to a malignant epigenetic landscape [189]. Among the various histone modifications, two have been particularly implicated in GBM pathogenesis: acetylation and methylation [190]. Notably, histone methylation has distinct implications in pediatric and adult GBM. The gene encoding the histone H3 isoform (H3F3A) frequently harbors somatic missense mutation. Indeed, H3K27M occurs in up to 80% of GBM or in diffuse intrinsic pontine glioma (DIPG) patients. In H3K27M mutant cells, a methionine is encoded instead of a lysine as the 27th amino acid on the N-terminal tail of the histone H3.3, resulting in an abnormal active chromatin status. H3K27M mutation has a crucial impact on the epigenetic landscape of glioma: (i) a wide decrease in H3 trimethylation (H3K27Mme3) resulting in chromatin de-condensation; (ii) a simultaneous increase in H3K27Mac, a mark recognized by BET proteins to recruit RNA polymerase II and activate transcription (Figure 2) [191,192]. Therefore, H3K27M mutation is involved in gliomagenesis by altering epigenetic control of gene expression via DNA methylation, chromatin structuring and deregulation of gene transcription pattern [193,194]. The loss of H3K27me3, which results in the formation of H3K27M-K27ac heterotypic nucleosomes, prompted Piunti and colleagues to investigate the association between acetyl-binding bromodomain proteins and H3K27M-containing nucleosomes (Figure 2). They found a significant overlap of BRD2 and BRD4 within the H3K27M-occupied sites [195]. In addition, they proposed that H3K27M-K27ac is involved in the formation of oncogenic super-enhancer-like elements, due to its abundant presence over a BRD2- and BRD4-highly occupied active enhancers fractions [196]. The strong co-occupancy of BRD2 and BRD4 members and H3K27M-K27ac in heterotypic nucleosomes is now considered evidence of BRD proteins involvement in DIPG pathogenesis [195,197]. Indeed, pharmacological inhibition of BET proteins via JQ1 treatment causes more effective anti-proliferative effects and neuron-like morphological changes in glioma H3K27M-mutant than in non-mutant GBM cell lines [195]. In GBM epigenome, the global genomic H3K27ac enrichment also enables BRD4 to regulate the immunophenotype of GSCs. BRD4, which preferentially binds H3K27ac [198], is thought to be responsible for the immunosuppressive transcriptome of Oct4/Sox2 co-expressing GSCs [199]. The progressive suppression of BET chromatin readers during the acquisition of a terminally differentiated phenotype of neural progenitors [154,155] and the discovery of BET overexpression in GBM tissues compared to normal brain [9,200,201] support an involvement of BET proteins in stemness maintenance. Regression to an undifferentiated stage and loss of cell identity leads GBM cells to resemble NSCs stem cells [202]. 

Over the past twelve years, an increasing body of knowledge has demonstrated that BET proteins play a pro-oncogenic role in GBM biology and have suggested that these epigenetic enzymes could be promising pharmacological targets [132,200,203,204,205,206,207]. However, until recently the molecular mechanisms by which BET inhibition could impair glioma survival in in vitro and in vivo models have remained poorly understood. Several studies are currently elucidating the signaling pathways regulated by the individual BET members and how their alteration may support GBM progression.

#### 5.2.1. BET Contribution to RTK/PI3K/AKT Signaling Pathway in Glioma

PI3K/AKT signaling pathway is frequently dysregulated in GBM cells, due to alterations in several signaling proteins. Two well-known GBM hallmarks are RTK amplification and phosphatase and tensin homolog (PTEN) loss of function or abnormal activation, which are an upstream and downstream component of PIK3/AKT axis, respectively [208]. Over the last few decades, hyper-activation of PI3K/AKT signaling pathway in malignant brain tumors, including GBM, has been frequently related to genetic and epigenetic alterations. Indeed, the role of BRD4 in PI3K/AKT signaling pathway [209] and its overexpression in GBM suggest a functional relationship. BRD4 silencing or pharmacological inhibition by JQ1 treatment seems to limit the stemness of murine GSCs and their cellular plasticity by decreasing the NSC marker Nestin and the ciliary neurotrophic factor. BRD4 inhibition impairs cell proliferation, induces apoptotic death by favoring DNA damage, and allows murine GSC to differentiate into astrocytes [206]. The main mechanism by which BET inhibition mediates these effects is through down-regulation of VEGF/PI3K/AKT pathway. BRD4 is known to regulate the key GSCs oncogenic properties by promoting VEGF expression; therefore, its knocking-down or inhibition reduces VEGF transcription and VEGFR activation and switches off all the downstream signaling pathways [206] (Figure 3).

#### 5.2.2. BET Role in Notch Signaling Pathway

The Notch signaling pathway is known to regulate several cellular processes, such as cell migration, differentiation, fate decision, apoptosis, stem cell maintenance, self-renewal and homeostasis, as well as neurogenesis and gliogenesis [210,211,212]. As a master regulator of NSCs commitment to a well-established cell fate at neurogenic niches [213,214], the Notch signaling pathway plays a key role in CNS during both the embryonic stage and in adulthood [215]. Indeed, Notch signaling is the main driver of NSCs specification process at the SVZ and at the DG in the adult hippocampus. Moreover, Notch expression levels have been shown to be critical for cell-fate decision: low Notch levels induce NSCs to undergo the last mitotic division before the exit from cell cycle and differentiation into neurons [216]; on the other hand, high Notch levels lead NSCs to growth arrest and to remain in a quiescent state [217,218]. Expression levels of Notch receptors (Notch 1 and 4) or Notch pathway components (DII1, DII4, Jagged 1, Hey1, Hey2, and Hes1) are higher in GBM tissue than in normal brain [219]. Particularly, Notch1, Notch2, and Notch4 expression levels are high in GSCs and their expression levels correlate with elevated stemness of the GBM cells [220,221,222], suggesting a contribution of the Notch signaling pathway in gliomagenesis. Downregulation of the Notch signaling pathway allows the awakening of the potential of latent neuronal differentiation of GSCs, resulting in a reduction in self-renewal and in the induction of neuronal differentiation [223]. 

Little is known about the epigenetic regulation of Notch signaling in GBM. It has been shown that HEY1, a component of the pathway is overexpressed in GBM compared to the healthy tissues due to an hypomethylation of CpG islands within the HEY1 promoter region [224]. Interestingly, after the first demonstration that BET family proteins, especially BRD4, affect self-renewal and tumorigenicity of breast cancer stem cells via the Notch signaling pathway [181], an analog mechanism was also observed in GSCs. BRD4 was found to be the major BET chromatin reader involved in the maintenance of GSCs stemness through the regulation of Notch1 axis [9]. BRD4 directly binds the Notch promoter and positively regulates Notch1 expression, thereby acting as a pro-tumorigenic factor that supports self-renewal (i.e., tumor proliferation and tumorspheres formation) and stemness maintenance [9] (Figure 3).

#### 5.2.3. BET Involvement in GLI1/IL6/STAT3 Signaling Axis

STAT3 is an inducible intracellular transcription factor involved in critical cellular processes, such as cell-cycle progression and anti-apoptotic program [225]. STAT3 activity is regulated by post-translational modification in response to interleukin 6 (IL6) and growth factors to regulate cell-cycle progression and/or apoptosis [226,227]. STAT3 is frequently found to be aberrantly active in several types of cancer [228,229], including GBM where it is also associated to a reduced patients’ survival [230]. BRD4 has been demonstrated to function as a molecular scaffold that enables the formation of the nuclear complex STAT3-CDK9 (cyclin-dependent kinase 9), which is required for RNA Pol II de-repression and activation, by a physical interaction with STAT3 following its NH2-terminal mono-ubiquitination [231,232]. 

The Hedgehog (HH) signaling pathway plays a critical role during embryogenesis, in cell proliferation, differentiation, stem cell maintenance, but is also involved in tumorigenesis [233,234]. GLI1 is a member of the GLI transcription factor family and is a main target gene of HH, acting as enhancer of the HH signaling pathway itself [235]. In GBM, GLI1 overexpression is caused by the hyperactivation of the HH pathway rather than by its genic amplification [236]. First in medulloblastoma and then in GBM, BRD4 was found to regulate GLI1 expression through direct binding to its promoter region [198,237]. Interestingly, in GSCs, BRD4 occupancy of GLI1 promoter is enriched in the presence of HDACi, because of the capability of BRD to bind the acetylated chromatin. A different and not fully understood interplay between GLI1 and STAT3 axis has also been shown, where GLI1 may positively regulate STAT3, promoting IL6 cytokine expression [198]. Nevertheless, the link between GLI1 and STAT3 pathway needs to be further investigated considering the impairment of GSCs survivability and their commitment to astrocytic lineage achieved by the combination of HDACs and BET protein inhibition [198] (Figure 3).

## 6. Reprogramming Therapy as GMB Treatment Strategy

During embryonic development, cells differentiate, assuming a well-defined identity and a specialized phenotype. As cells differentiate, there is a progressive restriction of the developmental plasticity coupled to cell cycle exit and a loss of proliferative and regenerative capacities. Oncogenic transformation reverses the differentiation stage and restores cells in a progenitor stem-like phenotype [21]. Currently, the reprogramming strategy is considered an attractive approach in cancer therapy due to the advantage of bypassing the conversion of tumor cells into progenitor cells. The control and the manipulation of cell fate in order to generate the desired cell type is an old goal that has been the subject of intense research. The limited regenerative ability of the CNS is encouraging the progress in cell replacement therapy and regenerative medicine, especially in the fields of neural injury and neurodegeneration [238]. The ectopic expression of transcription factors (TFs) was the first method used to reprogram neurons and neural progenitors both in vitro and in vivo [239,240,241,242,243,244,245,246,247]. Indeed, the sustained expression of well-known transcription factors allows a switch of non-neuronal cells (astrocytes or fibroblasts) into reprogrammed neurons with specific electrophysiological activity such as glutamatergic, GABAergic, and dopaminergic neurons, by modifying the established transcriptional program [240,244,248,249,250]. Manipulation of the TFs-mediated cellular phenotype opened the door to the “chemical reprogramming” leading to the identification of the main factors involved in transcriptional cascades, epigenetic modifications, and genetic networks [251,252,253,254]. The so-called Chemical transdifferentiation has allowed an improvement in the conversion efficiency [255,256,257,258,259], modulating chromatin accessibility or increasing TFs binding [260,261], offering advantages such as easier administration and druggability. Several small molecules cocktails have been empirically fine-tuned to directly convert human astrocytes into neurons in in vitro models [262,263]. Indeed, the exact signaling pathways that are upregulated or switched off during chemical transdifferentiation have only recently been elucidated [264]. Currently, a novel perspective of in vivo cell fate reprogramming is being successfully pursued, consisting of direct in situ chemical conversion of resident cells to compensate for cell loss and endogenous tissues repair [265]. However, the reprogramming of cancer cells into their healthy functional counterparts has lagged behind [266,267,268] due to the high heterogeneity among the specific epigenetic landscapes of cancers [266]. Indeed, the major hurdle in chemical reprogramming of malignant cells into terminally differentiated ones is understanding the molecular mechanisms that operate during the tumorigenesis and the specific individual cancer-initiating cell epigenome. Nevertheless, differentiation therapy based on all-trans-retinoic acid (ATRA) administration has shown significant clinical benefits for the treatment of hematological malignancies treatment, particularly for acute promyelocytic leukemia [269,270]. Clinical complete remission rates exceed 90% in patients with acute promyelocytic leukemia, after treatment with the differentiation inducing-agents ATRA and arsenic trioxide (As2O3), either alone or in combination [271]. However, this predominant differentiation-inducing activity has never been achieved in solid tumors. In GBM, direct reprogramming of proliferating cancer cells into post-mitotic neural cells is considered a more effective approach than most of the therapies aimed at killing them. To date, the most relevant results have been achieved by ectopic expression of specific TFs [272,273,274,275]. Interestingly, single TFs overexpression led to enrichment of GBM-reprogrammed neuronal cultures with neurons endowed with specific electrophysiological activities [276]. In detail, the sustained expression of Neurog2 or NeuroD1 promoted the transdifferentiation of GBM cells to forebrain glutamatergic neurons, whereas the ectopic expression of Ascl1 facilitated the acquisition of a GABAergic phenotype [276]. Chemical reprogramming did not always demonstrate the same efficiency in GBM transdifferentiation. A cocktail of small molecules was efficiently able to impair GBM tumorigenicity maintaining reprogrammed cells into an early metastable neuronal state, albeit not inducing cell terminal differentiation [184]. Instead, an FDA approved drug cocktail, consisting of HDAC, TGF-β, Rho Kinase, and GSK3β inhibitors, showed the ability to induce: (i) neuronal-like morphology, (ii) neuronal genes expression pattern, (iii) specific neuronal electrophysiological properties, (iv) attenuation of aggressiveness in GBM cell models, and suppression of tumor growth and decreased survival in GBM xenograft models in rodents [277]. A neuronal-like phenotype was also achieved in GBM cell models treated with protein kinase inhibitors (Rho-associated protein kinase, ROCK) and with mammalian target of rapamycin (mTOR) [278].

The key role of BET proteins in transcriptional regulation and cell fate decision makes them effective targets for reprogramming therapy. Several studies have shown that BET proteins inhibition alone significantly impairs GBM tumorigenicity both in vitro and in vivo, conferring a neuronal-like identity [9,195,198,206] and improving sensitivity to the first line chemotherapeutic drug TMZ [279,280,281]. Nevertheless, although BET proteins are considered promising pharmacological targets for brain diseases and for GBM [132,182], the lack of a selective inhibitor targeting the single members of the BET family has limited their progression in clinical trials so far. Research efforts are moving in order to identify molecules able to selectively recognize and inhibit BD1and/or BD2 of the individual BET reader and shoot down the side effects caused by the currently available pan-BET inhibitors. 

## 7. Concluding Remarks

BET proteins act as chromatin readers and transcriptional regulators. Because of their peculiar functions, BET members have been largely involved in the modulation of cell plasticity during mammalian embryogenesis. Indeed, depending on the lineage progenitor cell, BET proteins can promote both stemness maintenance and terminal differentiation. During carcinogenesis BETs sustain tumor growth and aggressiveness, thus playing a central role in cancer biology. There is increasing evidence that the epigenetic landscape is profoundly altered in GBM, leading to hyperacetylation and subsequent abnormal active chromatin status. Consistently, the recruitment of BET proteins is increased at promoter regions and enhancers of several genes, particularly those involved in survival, proliferation, and stemness maintenance. From these premises, it is clear that the activity of BET proteins is crucial for the loss of cell identity and aggressiveness in GMB, and its pharmacological targeting may represent a valuable therapeutic avenue in the near future. Reprogramming of malignant cells into healthy counterparts is indeed considered a promising approach in cancer and the so called “reprogramming therapy” is now emerging as an innovative strategy in the clinic. In this regard, the BET family represents a good candidate target to convert highly proliferating malignant cells into post-mitotic ones. Due to the compelling demand of effective therapeutic strategies, the design of efficient and safe molecules capable of inhibiting BETs in the cerebral tumoral tissue may represent a major challenge in the clinical management of GBM.

## Figures and Tables

**Figure 1 ijms-24-05665-f001:**
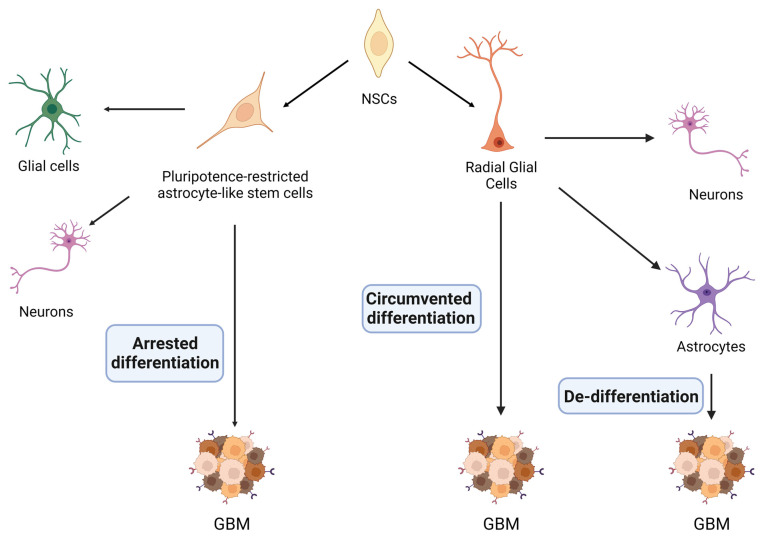
Putative mechanisms of GBM onset. During the embryonic development, neural stem cells (NSC) are able to differentiate in a broad phenotypic spectrum, gradually acquiring specialized cellular identities. Three main models have been proposed to explain the loss of cellular identity occurring during gliomagenesis: (i) De-differentiation model, in which de-differentiated cortical astrocytes represent the GBM cells-of-origin; (ii) Arrested Differentiation model, according to which the glioma-initiating cells are multipotent progenitor cells whose proper differentiation is blocked; (iii) Circumvented differentiation model, where a pool of quiescent embryonic progenitors which is dormient in the adult brain re-assume a proliferative expansion originating a tumoral mass.

**Figure 2 ijms-24-05665-f002:**
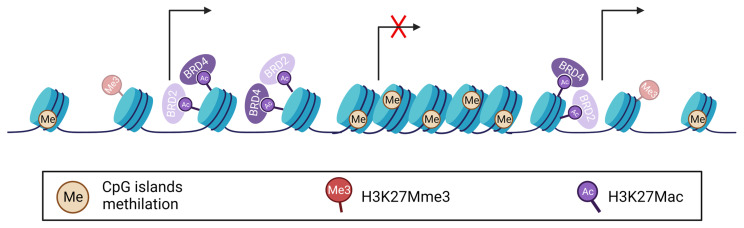
GBM Epigenome. H3K27M mutation results in global genomic decrease in H3 trimethylation (H3K27Mme3) and a concomitant increase in H3 acetylation (H3K27Mac), coupled to enhanced BRD2 and BRD4 recruitment and aberrant induction of gene expression. Tumor-suppressors hypermethylation also occurs.

**Figure 3 ijms-24-05665-f003:**
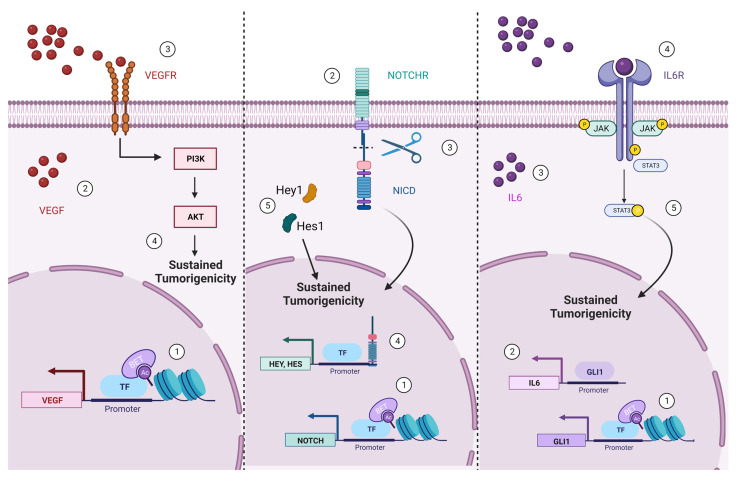
Oncogenic pathways regulated by BET proteins in GBM. BET proteins positively regulate some oncogenes involved in the main GBM dysregulated signaling pathways promoting survival, death evasion and stemness maintenance. The numbers in the caption indicate the temporal sequence of the events.

## Data Availability

Not applicable.

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
