# Peer review of "Regulation of Cell Plasticity by Bromodomain and Extraterminal Domain (BET) Proteins: A New Perspective in Glioblastoma Therapy"

_ijms, 2023, doi:10.3390/ijms24065665_

Round 1
Reviewer 1 Report
The review provides a comprehensive coverage of the current state of knowledge in the field of cancer epigenetics with a specific focus on the role of chromatin modifying proteins members of the Bromodomain and Extraterminal domain (BET) family in glioblastoma. The authors have elaborated in great detail the functional and mechanistic aspects of BET proteins known from neurogenesis and propose an integrated view of BET-driven transcriptional reprogramming of neural stem cells/progenitors as a driving force in the development of glioblastoma. Perspectives and challenges in the therapeutic targeting of BETs in glioblastoma are also addressed. The review is well structured and has a clear, logical flow.
On the downside, some statements are either arguable or not quite accurate and diminish the overall merit of the review.
Line 27: statement that “GBM is the most common and aggressive malignant primary brain tumor in human” reflects the situation in adults but not with pediatric brain tumors
Lines 31,32: “impossibility .. to remove all tumor cells” and “capability to invade the surrounding tissue“ are related to the same phenomenon, namely a highly infiltrative nature of GB.
Lines 100-102: Statement that “GSCs express typical stem cell biomarkers…CD133, Sox2 and Notch” is misleading because these markers are not universal and expressed in only some but not all GSC subtypes. In fact, the utility of CD133 as a reliable identity marker for GSCs has been questioned in the past decade with an array of studies reporting on GSCs lacking CD133.
Line 102: ability to “form tumors in orthotopic xenograft animal models “ is not a distinctive property of GSCs as differentiated gioma cells lacking stemness attributes also give rise to tumors in xenograft models.
Line 110: Statement that self-renewal is essential for the GSCs capacity to “sustain tumor mass growth” is highly debatable. While the propensity to self-renew is a fundamental attribute of stemness self-renewing GSCs proliferate slowly. Recent evidence indicates that in order to promote tumor growth GSCs actually need to exit from the slow proliferative mode associated with self-renewal.
Line 114: Statement that “GSCs …substantially contribute to tumor metastasis and recurrence“ is confusing because GBs are not metastasizing tumors and spread rarely outside the CNS.
Unclear or inadequate definitions is another drawback. To name just a few:
“restricted lifelong self-renewal potential” (line 161)
“GFAP expression is responsible for the astrocyte lineage commitment of GFAP-positive progenitors” (line 189)
„mitotic somal translocation…observed only during human development” (line 195)
“H3K27M encodes a methionine is encoded instead of a lysine as the 27th amino acid on the N-terminal tail of the 453 histone H3.3 …” (line 452)
Author Response
POINT-BY-POINT REBUTTAL LETTER
We would like to thank the Editor and the Reviewers for their work. Based on their useful comments, the review manuscript is substantially improved. We sincerely hope to have fully satisfied the Reviewer's requests and that the manuscript is now suitable for publication.
Reviewer #1 (Remarks to the Author):
The review provides a comprehensive coverage of the current state of knowledge in the field of cancer epigenetics with a specific focus on the role of chromatin modifying proteins members of the Bromodomain and Extraterminal domain (BET) family in glioblastoma. The authors have elaborated in great detail the functional and mechanistic aspects of BET proteins known from neurogenesis and propose an integrated view of BET-driven transcriptional reprogramming of neural stem cells/progenitors as a driving force in the development of glioblastoma. Perspectives and challenges in the therapeutic targeting of BETs in glioblastoma are also addressed. The review is well structured and has a clear, logical flow.
We thank the Reviewer for his/her nice comments on our manuscript.
On the downside, some statements are either arguable or not quite accurate and diminish the overall merit of the review.
Line 27: statement that “GBM is the most common and aggressive malignant primary brain tumor in human” reflects the situation in adults but not with pediatric brain tumors
The Reviewer is right and we properly corrected the sentence.
Lines 31,32: “impossibility .. to remove all tumor cells” and “capability to invade the surrounding tissue“ are related to the same phenomenon, namely a highly infiltrative nature of GB.
We modified the sentence, as suggested.
Lines 100-102: Statement that “GSCs express typical stem cell biomarkers…CD133, Sox2 and Notch” is misleading because these markers are not universal and expressed in only some but not all GSC subtypes. In fact, the utility of CD133 as a reliable identity marker for GSCs has been questioned in the past decade with an array of studies reporting on GSCs lacking CD133.
We modified the sentence.
Line 102: ability to “form tumors in orthotopic xenograft animal models “ is not a distinctive property of GSCs as differentiated gioma cells lacking stemness attributes also give rise to tumors in xenograft models.
The Reviewer is right in stating that non only GSCs cells, but also differentiated or partially differentiated glioma cells can give rise to tumors in xenografts models, but, in this sentence, we only describe some properties that GSCs share with NSCs, including the ability fo form tumors.
Line 110: Statement that self-renewal is essential for the GSCs capacity to “sustain tumor mass growth” is highly debatable. While the propensity to self-renew is a fundamental attribute of stemness self-renewing GSCs proliferate slowly. Recent evidence indicates that in order to promote tumor growth GSCs actually need to exit from the slow proliferative mode associated with self-renewal.
We thank the Reviewer for the clarification and we agree with him about the recent advances in understanding the physiology of GSCs that define them as slow cycling cells, almost in a quiescent state. Indeed, slow cycling GCSs are less sensible to DNA damaging cytotoxic drugs, which mainly target fast cycling cells. Therefore, GSCs survival leads to tumor recurrence. However, despite the use of inappropriate terms, with that statement we meant to emphasize the high degree of cell plasticity that allow GSCs to adopt a broad phenotypic spectrum and promote tumor heterogeneity. We changed our statement accordingly.
Line 114: Statement that “GSCs …substantially contribute to tumor metastasis and recurrence“ is confusing because GBs are not metastasizing tumors and spread rarely outside the CNS.
We agree with the Reviewer that the term "metastasis" is inappropriate to indicate the invasive capability of GBMs and we modified it in "spreading".
Unclear or inadequate definitions is another drawback. To name just a few:
“restricted lifelong self-renewal potential” (line 161)
“GFAP expression is responsible for the astrocyte lineage commitment of GFAP-positive progenitors” (line 189)
„mitotic somal translocation…observed only during human development” (line 195)
“H3K27M encodes a methionine is encoded instead of a lysine as the 27th amino acid on the N-terminal tail of the 453 histone H3.3 …” (line 452)
We thank the Reviewer for the accurate analysis of the text. We have amended it as suggested and carefully check the manuscript for spelling and grammar mistakes.
Reviewer 2 Report
It's a good review paper in general. The authors addressed the importance of BET bromodomains in glioblastoma. It's very organized and the subtitles are super helpful. The figures are very clear but the resolution are low.
There were a few spell check needed, for example: Line 211: BET proteins, Line 223: amino-terminal, Line 276: iBET, Line 337: A hallmark, Line 342: writers, etc.
There was a case the authors used Brd4(Line 400) instead of BRD4. They were different. Make sure to use the right one.
In 3.1. BET protein inhibitors, the author were trying to introduce the reported inhibitors and their potentials in treatment of diseases but the references were not sufficient to cover all the diseases. In addition, the authors mentioned about the lack of selective BET inhibitors (for single domain or pan-D1/D2) multiple times (Line 292, 317, 325, 371, 636, 655). There were a few reported selective inhibitors against BRD4 BD1, all BD1s, and all BD2s. It's better to include those inhibitors to make the story more comprehensive.
Line 325: "selectively inhibiting the two bromodomains of the 67 BET members." Do you mean 4 BET members?
Line 319: The authors were talking about PROTAC molecules as a new generation of iBETs for better selectivity. There was a reported a (+)-JQ1 based degrader which preferred to down-regulate BRD4 over other BET proteins. But I don't think all BET PROTACs have selectivity. In addition, reference 131 only talked about the E3 ligase but not related to BET bromodomains.
In section 5, the authors were talking about the function of BET bromodomains in tumor biology. They used a lot of examples to show the roles and BRD2, BRD3, and BRD4. Is there any report using selective BET inhibitors to better elucidate the role of each protein?
Author Response
POINT-BY-POINT REBUTTAL LETTER
We would like to thank the Editor and the Reviewers for their work. Based on their useful comments, the review manuscript is substantially improved. We sincerely hope to have fully satisfied the Reviewer's requests and that the manuscript is now suitable for publication.
Reviewer #2 (Remarks to the Author):
It's a good review paper in general. The authors addressed the importance of BET bromodomains in glioblastoma. It's very organized and the subtitles are super helpful. The figures are very clear but the resolution are low.
We thank the Reviewer for his/her nice comments on our manuscript. We provided high resolution figures in the revised manuscript.
There were a few spell check needed, for example: Line 211: BET proteins, Line 223: amino-terminal, Line 276: iBET, Line 337: A hallmark, Line 342: writers, etc.There was a case the authors used Brd4(Line 400) instead of BRD4. They were different. Make sure to use the right one.
We modified the text accorindgly to the Reviewer's comments.
In 3.1. BET protein inhibitors, the author were trying to introduce the reported inhibitors and their potentials in treatment of diseases but the references were not sufficient to cover all the diseases.
We agree with the Reviewer's comment and added some new References, accordingly.
In addition, the authors mentioned about the lack of selective BET inhibitors (for single domain or pan-D1/D2) multiple times (Line 292, 317, 325, 371, 636, 655). There were a few reported selective inhibitors against BRD4 BD1, all BD1s, and all BD2s. It's better to include those inhibitors to make the story more comprehensive.
We thank the Reviewer for his/her suggestion. Based on this valuable comment, we added a description of different new iBET molecules, including the more selective ones.
Line 325: "selectively inhibiting the two bromodomains of the 67 BET members." Do you mean 4 BET members?
We thank the Reviewer for highliting this editing mistake and eliminated that number.
Line 319: The authors were talking about PROTAC molecules as a new generation of iBETs for better selectivity. There was a reported a (+)-JQ1 based degrader which preferred to down-regulate BRD4 over other BET proteins. But I don't think all BET PROTACs have selectivity.
Based on this comment, we added some more infos about the PROTAC iBETs. As properly discussed by the Reviewer, not all PROTAC iBET have high selectivity, as we reported.
In addition, reference 131 only talked about the E3 ligase but not related to BET bromodomains.
We apologize for the mistake and we corrected the reference.
In section 5, the authors were talking about the function of BET bromodomains in tumor biology. They used a lot of examples to show the roles and BRD2, BRD3, and BRD4. Is there any report using selective BET inhibitors to better elucidate the role of each protein?
This is a very interesting point. As we discussed in the 5th paragraph, the main hurdle in reach a full comprehension of the role of the single BET members is the lack of specific inhibitors. Indeed, despite the identification of small molecules with higher affinity for BDs belonging to single BET members, until now all the available molecules show a minimal overlap between at least two members. We found only an evidence about the use of the bivalent BETi AZD5153, optimized to only interact with both BDs of BRD4 (new Ref. 134). However, to my knowledge, no evidence about the molecular effects on tumors biology of BRD4 selective inhibition are reported so far. Notably, a further obstacle in understanding the oncogenic role of BET protein in brain tumors is represented by the need to develop molecules able to cross the blood-brain barrier.